# Statins Directly Influence the Polarization of Adipose Tissue Macrophages: A Role in Chronic Inflammation

**DOI:** 10.3390/biomedicines9020211

**Published:** 2021-02-19

**Authors:** Sona Kauerova, Hana Bartuskova, Barbora Muffova, Libor Janousek, Jiri Fronek, Marek Petras, Rudolf Poledne, Ivana Kralova Lesna

**Affiliations:** 1Laboratory for Atherosclerosis Research, Centre for Experimental Medicine, Institute for Clinical and Experimental Medicine, 140 21 Prague, Czech Republic; kubh@ikem.cz (H.B.); rupo@ikem.cz (R.P.); ivka@ikem.cz (I.K.L.); 2Department of Physiology, Faculty of Science, Charles University, 128 00 Prague, Czech Republic; 3Department of Immunology, Faculty of Science, Charles University, 128 00 Prague, Czech Republic; barboramuffova@gmail.com; 4Department of Transplantation Surgery, Institute for Clinical and Experimental Medicine, 140 21 Prague, Czech Republic; lija@ikem.cz (L.J.); jifr@ikem.cz (J.F.); 5Department of Epidemiology and Biostatistics, Third Faculty of Medicine, Charles University, 100 00 Prague, Czech Republic; marek.petras@lf3.cuni.cz; 6Department of Anesthesia and Intensive Medicine, First Faculty of Medicine, Charles University and University Military Hospital, 169 02 Prague, Czech Republic

**Keywords:** statins, human, macrophage polarization, inflammation, hypercholesterolemia

## Abstract

Statins represent one of the most widely used classes of drugs in current medicine. In addition to a substantial decrease in atherogenic low density lipoprotein (LDL) particle concentrations, several large trials have documented their potent anti-inflammatory activity. Based on our preliminary data, we showed that statins are able to decrease the proportion of pro-inflammatory macrophages (CD14+16+CD36high) in visceral adipose tissue in humans. In the present study including 118 healthy individuals (living kidney donors), a very close relationship between the pro-inflammatory macrophage proportion and LDL cholesterol levels was found. This was confirmed after adjustment for the most important risk factors. The effect of statins on the proportion of pro-inflammatory macrophages was also confirmed in an experimental model of the Prague hereditary hypercholesterolemia rat. A direct anti-inflammatory effect of fluvastatin on human macrophage polarization in vitro was documented. Based on modifying the LDL cholesterol concentrations, statins are suggested to decrease the cholesterol inflow through the lipid raft of macrophages in adipose tissue and hypercholesterolemia to enhance the pro-inflammatory macrophage phenotype polarization. On the contrary, due to their opposite effect, statins respond with anti-inflammatory activity, affecting the whole organism.

## 1. Introduction

Coronary heart disease (CHD) mortality is decreasing worldwide (the trend started in the USA) [1,2]), with the main reason being favorable changes in risk factors [3]. These beneficial changes are responsible for about 50% of all morbidity changes in CHD mortality. Among all risk factors for CHD, the one with the biggest impact is decreased low-density lipoprotein (LDL) cholesterol levels [4]. Increased concentrations of LDL cholesterol and low concentrations of high-density lipoprotein (HDL) particles (mostly referred to as nascent HDL particles), together with the accelerated migration of immune cells—predominantly of monocytes—to the arterial wall, are involved in the early stages of atherogenesis [5,6]. Despite a substantial decrease in LDL cholesterol levels by statin therapy or dietary intervention, a risk remains for CHD in addition to cholesterol [7]: subclinical inflammation, with a crucial role played by adipose tissue and adipose tissue macrophage polarization [8]. Macrophage polarization in adipose tissue to pro-inflammatory phenotypes is associated with several general risk factors of cardiovascular diseases [9]. The combined effect of hypercholesterolemia and inflammation (as documented by the high-sensitivity C-reactive protein (hs-CRP)) was proved by the CHD mortality data [10,11,12,13]. The atherogenic effect of LDL cholesterol is counterbalanced by the protective role of HDL cholesterol, reducing the cholesterol content through reverse cholesterol transport [14]. The proportion of cholesterol influx versus efflux through the cell membrane affects the cholesterol content of lipid rafts and may influence macrophage polarization and inflammatory status regulation [14,15].

Although widely used in the treatment of hypercholesterolemia, statins have also been shown to possess anti-inflammatory activity [16,17,18,19,20,21]. This effect has been repeatedly observed in various population-based studies (The Cholesterol and Recurrent Events (CARE), pravastatin inflammation/CRP (PRINC) evaluation, and Justification for the Use of statins in Prevention: An Intervention Trial Evaluating Rosuvastatin (JUPITER)) where hs-CRP significantly decreased the independent level of LDL cholesterol [22,23,24]. Although the exact mechanism of the anti-inflammatory effect has not yet been elucidated, the acute administration of atorvastatin may prevent sepsis progression [25] in medical practice.

Recent data from in vitro and animal models have indicated that the macrophage phenotype might be directly influenced by statins [26,27]. Atorvastatin was shown to decrease the ability of macrophages to infiltrate into tissues, as documented by the decreased amount of filopodia formation and the ability of the extracellular matrix to degrade in macrophages polarized to pro-inflammatory macrophages in vitro [26]. This agrees with an in vivo animal model where atorvastatin reduced the macrophage infiltration into inflamed tissue [28]. In obese mice, statin therapy decreased the gene expression of pro-inflammatory cytokines [29].

In our previous studies, we successfully defined human adipose tissue macrophages based on the expression of selected surface markers (CD14, CD16, and CD36), i.e., pro-inflammatory macrophages (CD14+CD16+CD36high), transient macrophages (CD14+CD16+CD36low), and anti-inflammatory macrophages (CD14+CD16–CD36low) [9,30]. In our preliminary data analyzing macrophage polarization in adipose tissue, a surprisingly significant effect of statin was found. However, macrophage polarization is defined not only by changes in the expression of surface markers but, also, by the production of various cytokines and that of low-molecular weight substances (nitric oxide (NO)). The aim of this study was to investigate this effect of statins on macrophage polarization in a much larger number of individuals, combining data from two studies together with an animal model, and the in vitro statin effect.

## 2. Methods

The presented results combined data obtained from two subsequently enrolled groups of living kidney donors (the first from 2013 to 2015 and the second from 2017 to 2019). Since the analysis methods were essentially identical in both projects, it was then possible to combine the data. The only difference was the type of flow cytometry analyzer, as mentioned below. The design of our both studies was approved by the Ethics Committee of the Institute for Clinical and Experimental Medicine and Thomayer Hospital, Prague, Czech Republic and complied with the Declaration of Helsinki (27/June/2012, 1041/12 and 8.6.2016, G-16-06-22). The clinical data were collected from the clinical documentation of the enrolled subjects and from an interview targeting lifestyle factors. All participants were fully informed about the study and signed informed consent prior to participation.

### 2.1. Adipose Tissue Specimen

Visceral adipose tissue (VAT) was obtained during the cleansing of an isolated kidney (from the area outside of Gerota’s fascia) intraoperatively during living kidney donor (LKD) nephrectomy. Our study pooled data from two separate analyses where sample recruitment was conducted during two periods: one between 2013 and 2015 (*n* = 52) and the other between 2017 and 2019 (*n* = 66). Visceral adipose tissue samples were processed immediately, as described earlier [30]. Briefly, samples were cleaned from all visible blood vessels, fibrous tissue, and residual blood, cut into approximately 2 mm pieces, and digested in collagenase II solution (2 mg/mL, Sigma-Aldrich, St. Louis, MO, USA) and bovine serum albumin (0.02 g/mL, Sigma-Aldrich, St. Louis, MO, USA) in phosphate-buffered saline (PBS) without calcium and magnesium (Biosera, Manila, The Philippines) for 15 min (37 °C). Digested samples were immediately cooled on ice and filtered through 150 μL and 50 µm strainers CellTrics (Sysmex, Nordstedt, Germany) and centrifuged.

### 2.2. Flow Cytometry

The macrophage subpopulations in the stromal vascular fraction (SVF) samples isolated using this technique were determined by flow cytometry (a CyAn flow cytometry analyzer for the group of patients recruited in the 2013–2015 period or a CytoFlex analyzer for patient samples recruited in the 2017–2019 period; Beckman Coulter, Brea, CA, USA). Monoclonal antibodies and fluorochromes (CD14-phycoerythrin-cyanine 7 (PC7), CD16-phycoerythrin-Texas Red-X, ECD, CD 36-fluorescein isothiocyanate, FITC, and Fixable Viability Dye eFluor™ 780; Thermo Fisher, Waltham, MA, USA) were used for the specific individual subsets of macrophages. The gating strategy is shown in Figure 1. The macrophages were defined as CD14+ cells, with more than 95% of these cells also expressing the CD36 scavenger receptor. Based on the expression of surface markers, we defined the following three main adipose tissue macrophage subpopulations: pro-inflammatory (CD14+CD16+CD36high), transient (CD14+CD16+CD36low), and anti-inflammatory (CD14+CD16-CD36low).

### 2.3. Lipoprotein Analysis

Blood samples (drawn from LKDs after an overnight fast) were obtained immediately before surgery, prior to anesthesia induction. Cholesterol was determined using an enzymatic method (Hoffmann-LaRoche, Basel, Switzerland). The high-sensitivity C-reactive protein (hs-CRP) was measured using an immunoturbidimetric assay with a Cobas Mira Plus autoanalyzer (Hoffmann-LaRoche, Basel, Switzerland). Subjects were arbitrarily considered hypercholesteremic if their fasting plasma cholesterol levels exceeded 5 mmol/L or if they were on a statin treatment. No other hypolipidaemic treatment was used by the subjects. The high-density lipoprotein (HDL) cholesterol concentration was measured after the phosphotungstate precipitation of the apolipoprotein-B-containing lipoproteins and LDL cholesterol levels were calculated.

In our animal experiment, cholesterol was measured in the blood plasma and in the lipoprotein particles (very low-density lipoprotein (VLDL) d < 1.006 g/mL, intermediate-density lipoprotein (IDL) d = 1.006–1.019 g/mL, LDL d = 1.019–1.063 g/mL, and HDL d = 1.063–1.210 g/mL) separated by sequential ultracentrifugation.

### 2.4. Animal Experimental Model with Prague Hereditary Hypercholesterolemic Rat

The Prague hereditary hypercholesterolemic (PHHC) rats [31] were housed in ventilated cages under a 12 h light cycle and provided free access to food and water. Rats (*n* = 16) were fed a high-cholesterol diet (1% cholesterol dissolved in 5% beef lard by weight was added to the standard diet) from 8 weeks of age for 19 weeks. Half of the group on the high-cholesterol diet (*n* = 8, statin treatment group) received simvastatin (5 mg/kg body weight/day, *n* = 8) for the last 5 weeks. The control group (*n* = 7) received the standard rat diet (Velaz, Prague, Czech Republic).

### 2.5. Flow Cytometry of Animal Adipose Tissue Samples

Animals were decapitated, heparinized blood was obtained, and perirenal adipose tissue was dissected from the left kidney and immediately processed. Cleaned adipose tissue (1 g) was processed in a solution of collagenase (2 mg/mL) in PBS with 2% bovine serum albumin. The released stromal vascular fraction (SVF) was strained and centrifuged in the same way as the human samples of adipose tissue. Macrophages in the SVF were analyzed by flow cytometry. Monoclonal antibodies and fluorochromes were used (CD11b, CD86, CD163, and Fixable Viability Dye eFluor™ 780).

### 2.6. Nitric Oxide Production by Macrophages In Vitro

Peripheral blood mononuclear cells were isolated from blood using Ficoll-Plaque PLUS (GE Healthcare, Chicago, IL, USA) according to the manufacturer’s instructions. The blood was collected from one donor. Next, CD14+ blood monocytes were isolated using an M-pluriBead^®^ Maxi Reagent Kit and CD14 M-pluriBead^®^ anti-human (Pluriselect, Leipzig, Germany) from peripheral blood mononuclear cells according to the manufacturer’s instructions. Isolated monocytes were subsequently plated and differentiated by macrophage colony-stimulating factor (Preprotech, Hamburg, Germany, 80 ng/mL) in RPMI-1640 with glutamine (Biosera, Manila, The Philippines), penicillin/streptomycin (Biosera, Manila, The Philippines), and 10% fetal bovine serum (Biosera) for 4 days. The differentiated adhered macrophages were stimulated for 48 h by a combination of lipopolysaccharide (LPS; Preprotech, Hamburg, Germany, 100 ng/mL) and interferon-gamma (IFN-γ; Gibco, Gaithersburg, MD, USA, 20 ng/mL) into pro-inflammatory M1 macrophages with or without the addition of fluvastatin (10 μmol/L, Sigma-Aldrich). In addition, anti-inflammatory M2 macrophages were prepared by stimulation with interleukin 4 (IL-4; Preprotech, Hamburg, Germany, 20 ng/mL) and IL-13 (Preprotech, Hamburg, Germany, 20 ng/mL) with or without the addition of fluvastatin (10 μmol/L, Sigma-Aldrich). Negative control macrophages were not stimulated by additional cytokines. After 48 h of incubation, the medium was collected and centrifuged (300× *g*, 5 min), and the supernatants were used for the nitric oxide assay.

The supernatants were incubated with nitrate reductase from the mold *Aspergillus Niger* (50 mU/mL, Sigma-Aldrich, St. Louis, MO, USA) and NADPH (final concentration = 100 μmol/L, Sigma-Aldrich, St. Louis, MO, USA) for 30 min at room temperature. After the reduction, the samples were incubated with a solution of methanol and diethyl ether (a 3:1 mixture, *v*/*v*). After a 1-h incubation at 4 °C, the samples were centrifuged (10,000× *g*, 10 min at 4 °C), and the supernatants were used for nitrite determination by the Griess reaction. A mixture of solution A (0.1% N-1-napthylethylenediamine dihydrochloride in water) and solution B (1% sulfanilamide in 5% H_3_PO_4_, 1:1, *v*/*v*) was added to the prepared samples. Absorbance was measured at 540 nm using a microplate reader (Multi-Detection Microplate Reader, Synergy 2, BioTek, Bad Friedrichshall, Germany). A standard curve was generated with sodium nitrite at concentrations ranging from 2.5 to 150 μmol/L.

### 2.7. Statistical Analysis

Data are presented as means with SDs for continuous variables or percentages. Intergroup comparisons of continuous variables and multiple linear regression adjustments were performed using one-way ANOVA complemented by Bonferroni’s multiple comparison test. A correlation analysis was performed using biostatistical GraphPad Prism software (GraphPad Software Inc., San Diego, CA, USA). In all tests, *p*-values higher than 0.05 were considered statistically nonsignificant.

The subjects were stratified according to the presence (strata 1) or absence (strata 0) of the justified predictor. The crude differences in macrophage phenotypes, including 95% confidence intervals between subjects with and without the predictor (i.e., strata 1 or strata 0), were estimated from the normal distribution.

The differences mutually adjusted for all considered predictors were obtained from multivariate linear normal regressions with Bayesian estimation using the Metropolis-Hastings algorithm. This method allowed the same inference for small samples as for large samples. The strict criteria with differences, expressed as a mean and 99% highest posterior density interval, were applied.

The association of macrophage subsets with the justified cardiovascular disease predictor was assessed pursuant to the inferiority and superiority achievements. If an adjusted difference in the macrophages subsets including a 99% credible interval was lower or higher than the delta margin, the predictor association was proven. Otherwise, if the inferiority or superiority of the macrophages subsets in strata 1 against strata 0 was not achieved, no predictor association was demonstrated. The clinically relevant delta margin was defined as −1.5% for evidence of inferiority and +1.5% for evidence of superiority.

A proportion of the macrophage subsets in the population with low cardiovascular diseases risk, further called the zero population (i.e., nonsmoking women, younger than 50 years, lower body fat, and without hypercholesterolemia), were obtained from a mathematical model whose eligibility was verified in a small set of subjects meeting the criteria for the zero population. Hypertension could not be used for the analysis due to a high autocorrelation with the hypercholesterolemia parameter.

## 3. Results

From the results of the combined data obtained from two subsequently enrolled groups of living kidney donors (LKDs, *n* = 118), 11 (9.3%) subjects received statin therapy and 44 (37.61%) subjects had hypercholesterolemia, defined as a total cholesterol > 5 mmol/L, and/or the use of statins. No subject received any other type of hypolipidaemic treatment. These groups did not statistically differ in any parameter except for age (first group: 47.36 ± 10.85 years, *n* = 53 and second group: 52.68 ± 11.44 years, *n* = 65; *p* < 0.05). Detailed demographic and subject characteristics are provided in Table 1. The group of LKDs represented the healthy Czech population, as they were slightly better compared to the 1% representative Czech population sample [32].

The increased proportions of the pro-inflammatory macrophages (CD14+CD16+CD36high) in visceral adipose tissue (VAT) correlated significantly with the increased LDL cholesterol levels (Figure 2). Similarly, the proportion of anti-inflammatory macrophages (CD14+CD16-CD36low) highly significantly decreased with the increasing LDL. No relationship with the transient phenotype (CD14+CD16+CD36low) was found.

The Bayesian analysis of macrophages in human VAT of LKDs confirmed that hyperlipidaemia leads to a significant increase in the proportion of macrophages undergoing pro-inflammatory polarization and a significant decrease in the proportion of macrophages undergoing anti-inflammatory polarization in VAT compared to the zero population (Figure 3A). In this analysis, hypercholesterolemia also slightly increased the proportion of the transient phenotype compared to the zero population in VAT. Statin therapy was associated with the anti-inflammatory effect of VAT macrophage polarization (Figure 3B). Statin therapy significantly decreased the proportion of pro-inflammatory macrophages compared to the zero population. The proportion of anti-inflammatory macrophages in VAT significantly increased with statin therapy compared to the zero population, whereas the transient phenotype population was not influenced by statin therapy.

### 3.1. Animal Experiments

The high-cholesterol diet significantly increased the total cholesterol concentration compared to the control (Figure 4). The analysis of the cholesterol concentrations in the lipoprotein fractions in our animal model (Prague hereditary hypercholesterolaemic (PHHC) rats) showed that the high-cholesterol diet (HCD) also significantly raised the cholesterol contents in the VLDL, IDL, and LDL compared to the control group. The high-cholesterol diet had no significant influence on the HDL cholesterol concentration compared to the control.

The statin administration significantly attenuated the effect of the high-cholesterol diet on the total cholesterol concertation (Figure 4). Similarly, statin administration significantly decreased the cholesterol concentrations in all the Apo protein B-containing lipoprotein fractions (very low-density lipoprotein (VLDL), intermediate-density lipoprotein (IDL), and LDL) compared to the high-cholesterol diet group. Surprisingly, statin therapy also increased the HDL cholesterol compared to both the control and high-cholesterol diet groups. In this context, there are substantial differences in the compositions of the lipoprotein fractions between rats and humans. Generally, in rats—in general, as well as in the PHHC strain—the HDL:LDL ratio is physiologically much higher compared to that in human plasma.

We observed an upward trend in the number of adipose tissue macrophages and the number of pro-inflammatory macrophages (CD11b+CD86+) per gram of adipose tissue from the control group (*n* = 7) to the statin therapy group (*n* = 8) and to the high-cholesterol diet group (*n* = 8) (Figure 5). The only significant difference was noted between the two extreme (control and high-cholesterol diet) groups (*p* < 0.05). To assess the effect of statin, the proportion of pro-inflammatory macrophages was more meaningful. Statin therapy significantly reduced the macrophage polarization to the pro-inflammatory phenotype compared to both the control (*p* < 0.01) and high-cholesterol diet (*p* < 0.01) groups. To our surprise, the high-cholesterol diet did not increase the proportion of pro-inflammatory macrophages compared to the controls, probably due to the relatively small differences in the LDL cholesterol concentrations (and adequate inflow of cholesterol to the cell membrane).

### 3.2. In Vitro Experiment

Any effect on the macrophages in vitro should be assessed after their adequate stimulation to the M1 phenotype (stimulated by lipopolysaccharide and interferon-γ) or M2 (stimulated by IL-4 and IL-13) (Figure 6). Compared to the negative control, macrophages undergoing pro-inflammatory polarization (M1) significantly enhanced the nitric oxide (NO) synthase (NOS) activity (*p* < 0.001), whereas no effect of the anti-inflammatory polarization on the NOS activity compared to the negative control was found.

Statin therapy significantly decreased the activity of NOS by M1 macrophages (*p* < 0.001), thus documenting its anti-inflammatory effect. Statin had no effect on the NOS activity of the M2 macrophages and the negative control.

## 4. Discussion

Our data clearly demonstrated a strong anti-inflammatory effect of statin therapy on the polarization of adipose tissue macrophages in both humans and rats. Statin therapy significantly increased the proportion of macrophages becoming anti-inflammatory while decreasing the proportion of macrophages polarized to pro-inflammatory ones in the VAT of healthy individuals. The anti-inflammatory effect of statins on the macrophage polarization was confirmed both in vitro and in an animal experiment.

Cholesterol plays a crucial role in the cell membrane structure and function. Recently, the intensive exchange of cholesterol molecules on the surface of red blood cells was described in detail [33]. Although the proportion of circulating monocytes is smaller compared to red blood cells, the inflow and outflow of LDL cholesterol was proven [33]. Increased cholesterol commutation in circulating monocytes drives them to macrophage maturation [33]. Similar changes in macrophage cells residing in adipose tissue might be expected.

The two main atherogenic effects of enlarged ectopic adipose tissue are dyslipidaemia and a chronic pro-inflammatory status. As shown by our earlier data [34], these factors are interlinked. The cholesterol molecule plays a multifunctional role in atherogenesis [35] and an important role in signal transduction [36]. The increased mole percentage of the cholesterol content in the raft might be able to change the signaling of monocytes in the arterial wall and the maturation of the residential macrophages [37,38]. The participation of the membrane raft structures in the induction of inflammation has been documented in previous experiments [35,39,40]. As such, cholesterol may play a substantial role in transmembrane signaling and cell differentiation [37].

Macrophage polarization in response to environmental factors leads to a highly variable tissue macrophage phenotype. Our data (Figure 3) from the Bayesian analysis showed that hypercholesterolemia drives the macrophage phenotype into a pro-inflammatory state. This agrees with the findings of a close correlation between the LDL cholesterol concentration and the proportion of pro-inflammatory macrophages (Figure 2). Pro-inflammatory macrophages are an important source of inflammatory agents in adipose tissue and drivers of chronic adipose tissue inflammation [39]. Less is known about the link between the cholesterol content in the cell membrane and macrophage polarization. However, previous findings have suggested that the cholesterol contents of lipid rafts play an important role in pro-inflammatory macrophage polarization [36] and the production of pro-inflammatory cytokines [40]. Increased cholesterol levels in the cell membrane increase the pro-inflammatory status of macrophages. Cholesterol loading of the lipid raft accelerates tumor necrosis factor alpha (TNF-α) signaling in the plasma membrane macrophage [40]; it is a case of cholesterol imbalance between the inflow and outflow of cholesterol from the macrophage raft. Moreover, the molecular proportion of free cholesterol was recently shown to be three times higher in the outer layer compared to the inner layer, and the gradient was suggested to be responsible for information transfer [41].

Several anti-inflammatory effects of statins have been demonstrated to date. Statins reduce the expression of adhesion molecules, a finding that agrees with the reduced relative cell adhesion and the reduced expression of pro-inflammatory chemokines [42]. Additional data from in vitro experiments showed a significant reduction in matrix metalloprotease activity and the loss of filopodium formation in macrophages stimulated to become pro-inflammatory [26]. This agrees with the results of the effect of atorvastatin therapy on the fibrous cap thickness (EASY-FIT; in coronary atherosclerotic plaque, as assessed by optical coherence tomography) study documenting an in vivo increased fibrous cup thickness and atherosclerotic plaque stabilization through statin therapy [43].

A possible explanation of the direct anti-inflammatory effect of statins is the decreased cholesterol content, even in the cell membrane, consequently influencing the membrane lipid raft formation and its impact on cell signaling and polarization. As described above, rafts are domains in the cell membrane with a high content of cholesterol [37]; hence, changes in cholesterol concentrations may influence the raft functions. Similar to the present in vivo analysis of human adipose tissue macrophages in healthy individuals, animal experiments have shown a trend toward the anti-inflammatory effects of statin therapy. It is rather difficult, if not impossible, to extrapolate experimental data obtained in animal models to human pathology. We can only assume that the basic and widely used human macrophage markers (CD14 and CD16) correspond to the surface markers of the rat macrophages CD11b, CD86, and CD163 [44]. In our animal model, statin therapy significantly reduced the effect of the high-cholesterol diet on the proportion of pro-inflammatory macrophages in the visceral adipose tissue. Likewise, statin therapy significantly decreased the blood cholesterol concentrations compared to the high-cholesterol diet group. There were decreases in the cholesterol content in the VLDL, IDL, and LDL, while the HDL cholesterol concentrations increased compared to the high-cholesterol diet group. We highlighted the differences in the lipoprotein particles between humans and our genetic rat model. In human blood, the LDL concentrations are higher than the HDL concentrations, whereas the proportion of LDL in rats is physiologically very low, as evident in our control group.

A direct effect of a statin (fluvastatin, the only water-soluble statin) was documented in an experiment using human monocytes in vitro. The rate of production of NO (i.e., NOS activity) was used to assess the shift in the stimulated macrophages to the pro-inflammatory phenotype (usually named M1). Fluvastatin significantly inhibited the NO production of the stimulated macrophages, whereas no effect on the anti-inflammatory macrophages was found. Statins not only decrease the LDL particle concentrations but also directly affect the anti-inflammatory changes in macrophages.

Our data also agreed with the recent results reported by Zhang et al. [45]. They found a decreased ratio of pro-inflammatory-to-anti-inflammatory macrophage phenotypes in atherosclerotic plaques following statin therapy, despite an increase in the serum cholesterol levels in this experimental model. Similarly, Wang et al. found a significant effect of simvastatin on anti-inflammatory macrophage polarization in vivo in a rat model [46].

## 5. Study Limitations

The first limitation is the pooled data of our two groups of LKDs being enrolled in two different periods and their adipose tissue samples being examined using two different flow cytometers. The two groups were slightly different in their mean anthropometrical parameters. The more recently enrolled group was slightly older, and the proportion of smokers and the male-to-female ratio were lower compared to the first group. Despite the limitations due to differences in age, the proportions of the macrophage phenotypes were comparable, allowing us to pool the data of both groups into a large group to better assess its impact.

A second limitation was the animal experiment. There is a possibility that the short dietary intervention and statin therapy somewhat biased our results. A prolonged intervention might have led to a stronger effect, as observed by Zhang et al., where the animals were treated for 27 weeks [45].

In conclusion, we documented the anti-inflammatory effects of statins resulting in pro-inflammatory macrophage polarizations in adipose tissues. A similar effect was confirmed in our experimental model with hypercholesterolaemic rats. This effect is direct, as we found a direct effect of statins on the macrophage polarization in vitro. The direct effects of intra-vasal concentrations of cholesterol and its raft cell membrane content were recently documented in red blood cells [33]. We can expect a similar scenario in circulating monocytes, as well as in residential macrophages in adipose tissue. If so, then the anti-inflammatory effects of the statins documented in our three different experimental approaches are probably realized through raft cholesterol.

## Figures and Tables

**Figure 1 biomedicines-09-00211-f001:**
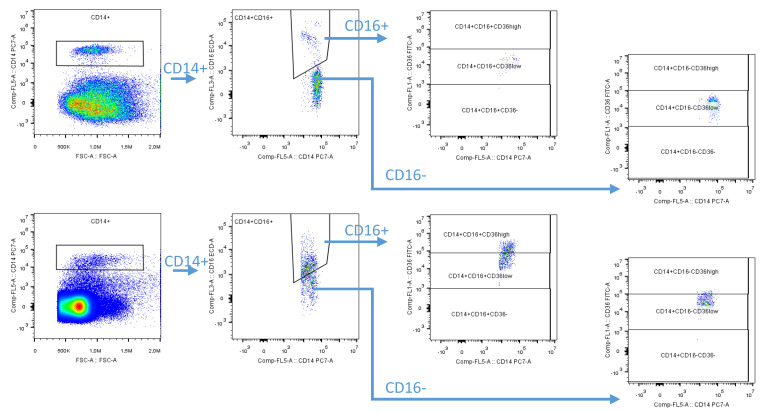
Blood gating strategy and the isolated stromal vascular fraction of the visceral adipose tissue. After gate-out debris and non-singlets, only viable cells were included in the analysis. Viable CD14+ cells were divided into CD16+ and CD16- subpopulations based on the blood sample gate where the transition between the CD16+ and CD16- population was obvious. Next, both cell populations were further differentiated according to CD36 expression into CD16-, CD36low, and CD36high. In this manner, we defined the three main adipose tissue macrophage subtypes as follows: pro-inflammatory (CD14+CD16+CD36high), transient (CD14+CD16+CD36low), and anti-inflammatory (CD14+CD16-CD36low).

**Figure 2 biomedicines-09-00211-f002:**
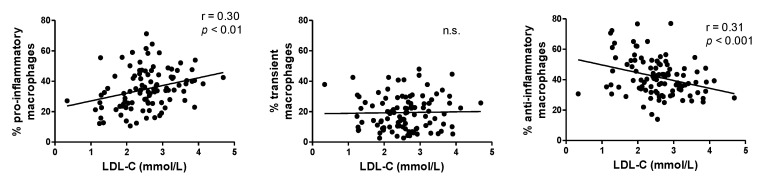
Polarization of adipose tissue macrophages and its relationship to low-density lipoprotein (LDL) cholesterol. Linear regression between LDL cholesterol and the percentage of pro-inflammatory (CD14+CD16+CD36high), transient (CD14+CD16+CD36low), and anti-inflammatory (CD14+CD16-CD36low) macrophages in visceral adipose tissue (VAT).

**Figure 3 biomedicines-09-00211-f003:**
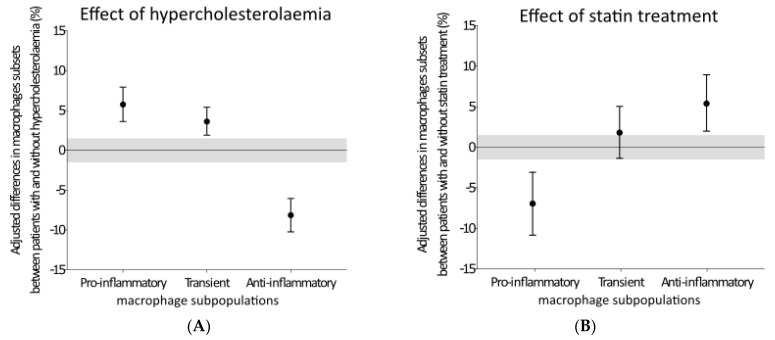
Effects of hypercholesterolemia (**A**) and statin therapy (**B**) in the 118 living kidney donors (LKDs) included in the Bayesian analysis. The data are expressed as the difference in the relative numbers of pro-inflammatory, transient, and anti-inflammatory phenotypes in VAT. The results are shown as the mean ± 95% confidential interval pro-inflammatory (CD14+CD16+CD36high), transient (CD14+CD16+CD36low), and anti-inflammatory (CD14+CD16-CD36low) macrophage subpopulations. The clinically irrelevant ±1.5% delta margins of the differences are shown as grey areas. The differences were obtained from multivariate linear normal regressions with Bayesian estimations using the Metropolis-Hastings algorithm.

**Figure 4 biomedicines-09-00211-f004:**
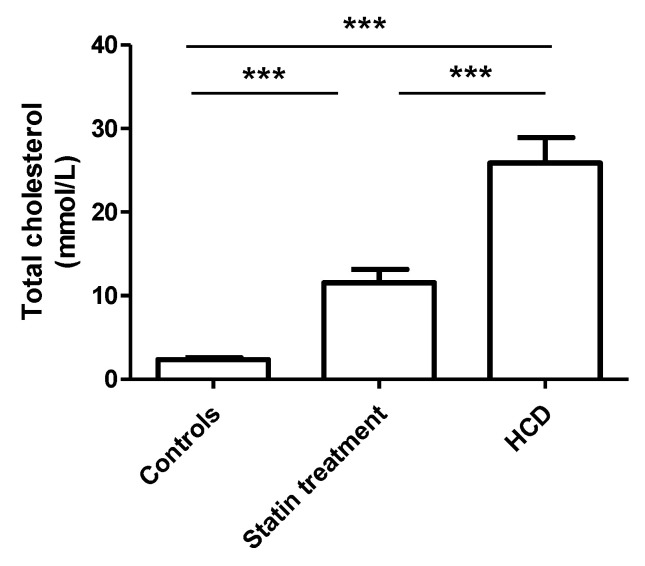
Total cholesterol plasma concentration of the rat. Control group (*n* = 7), statin therapy (*n* = 8), and high-cholesterol diet (HCD; *n* = 8). *** *p* < 0.001.

**Figure 5 biomedicines-09-00211-f005:**
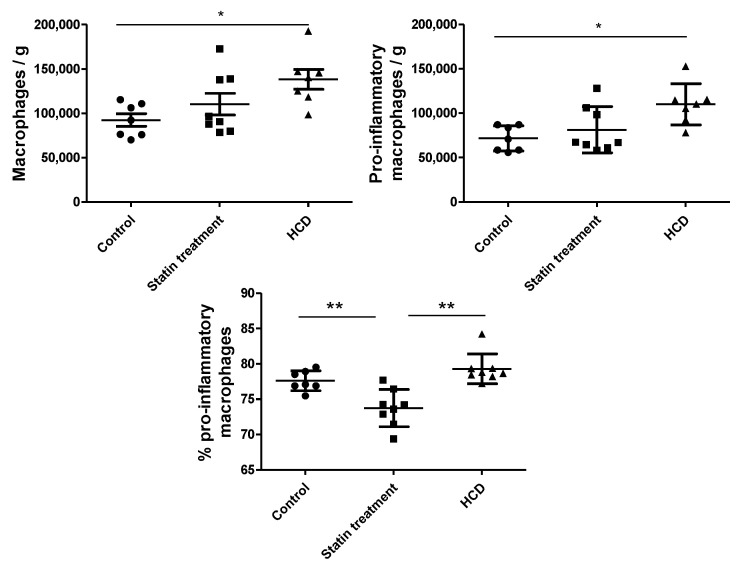
Numbers of macrophages (viable CD11b+ cells) and pro-inflammatory macrophages (CD11b+CD86+) per gram of rat adipose tissue and the proportion of pro-inflammatory macrophages of all the macrophages (CD11b+CD86+) in the control group (*n* = 7), statin therapy (*n* = 8), and HCD, high-cholesterol diet (*n* = 8); data are expressed as the number of cells per gram and proportion (%) in means ± SD. * *p* < 0.05 and ** *p* < 0.01.

**Figure 6 biomedicines-09-00211-f006:**
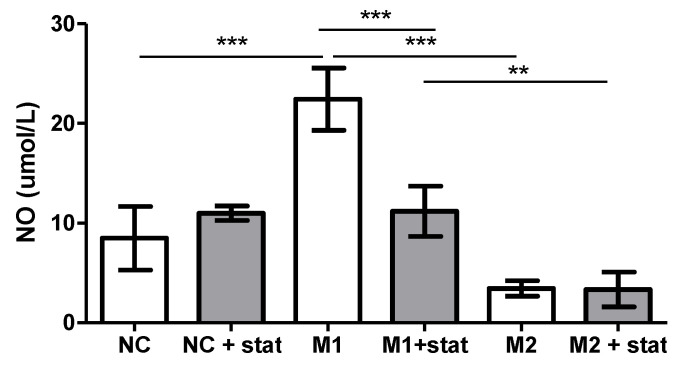
Concentrations of nitric oxide released into the medium by stimulated monocytes with or without statin. NC, negative controls, NC + stat, statin therapy for 48 h, M1, monocyte stimulation by lipopolysaccharides (LPS) and interferon-gamma (IFN-γ) for 48 h, M1 + stat, monocyte stimulation by LPS and IFN-γ and statin therapy for 48 h, M2, monocyte stimulation by interleukin 4 (IL-4) and IL-13 for 48 h, and M2 + stat, monocyte stimulation by IL-4 and IL-13 and statin therapy for 48 h. Each bar expresses the results of 4 wells in mean ± SD. ** *p* < 0.01 and *** *p* < 0.001.

**Table 1 biomedicines-09-00211-t001:** Characteristics of all living kidney donors (LKDs) included in the study (*n* = 118). n.s., nonsignificant, BMI, body mass index, HDL, high-density lipoprotein, LDL, low-density lipoprotein, and hs-CRP, high-sensitivity C-reactive protein.

Clinical and Biochemical Characteristics	*N* = 118	Subjects with Statins Treatment (*n* = 11)	Subjects without Statins Treatment (*n* = 107)	*p* with/without Statins Treatment
Age (years)	50.27 ± 12.3	60.03 ± 8.9	49.36 ± 11.26	*p* < 0.01
BMI (kg/m^2^)	25.59 ± 4.74	25.59 ± 2.85	25.55 ± 4.26	n.s.
Sex, male% (n)	30.5% (36)	18.2% (2)	31.8% (34)	n.s.
Hypertension% (n)	22.9% (27)	36.4% (4)	21.5% (23)	n.s.
Smoking% (n)	26.3% (31)	18.2% (2)	27.1% (29)	n.s.
Total cholesterol (mmol/L)	4.61 ± 0.99	3.95 ± 0.92	4.66 ± 0.87	*p* < 0.05
Total triglycerides (mmol/L)	1.53 ± 0.84	1.30 ± 0.65	1.55 ± 0.85	n.s.
HDL cholesterol (mmol/L)	1.31 ± 0.42	1.26 ± 0.46	1.32 ± 0.39	n.s.
LDL cholesterol (calculated) (mmol/L)	2.55 ± 0.79	2.10 ± 0.75	2.54 ± 0.94	n.s.
hs-CRP (mg/L)	1.40 ± 2.47	0.85 ± 0.60	1.46 ± 2.59	n.s.

## Data Availability

Data presented in this study are available on request from corresponding author. Data are not publically available due to privacy and ethical restrictions.

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
