# Peer review of "Statins Directly Influence the Polarization of Adipose Tissue Macrophages: A Role in Chronic Inflammation"

_biomedicines, 2021, doi:10.3390/biomedicines9020211_

Round 1
Reviewer 1 Report
In the "results" you did not describe clearly subjects of investigated population of 118 living kidney donors.
You mentioned about two separated groups (taken from different periods). You did it almost on the end of the paper because you started with the "results" just after "introduction"
You did not present these groups in results. "Materials and methods " ought to be earlier then "results", for better comprehension.
I propose you the order: Introduction, Materials and Methods, Results, Discussion, and Conclusions (optional).
You did not specify clearly the investigated subgroups, e.g. (line 84-86) " 11 (9.3%) subjects receiving statin therapy and 44 (37.61%) subjects with hypercholesterolemia defined as total cholesterol > 5 mmol/l and/or use of lipid lowering agents" - which one? and moreover define clearly please, the sort of groups. How to understand these numbers 11+ 44 ? - it does not give us 118. What about the rest of subjects?
Do not consider the well known facts like the relation between statins and lipids in the "chapter" presenting "animal experiments" - 123-146. Remove this part, please. Be focused on macrophages, please. The anti-inflammatory effect of statins is also well known but still is not highlighted adequately.
Most probably you added this part of paper (animal experiments) due to the results concerning macrophages in rats. Your paper has a value because of the anti-inflammatory effect of statins on macrophages not because you analyzed something (lipids) which had been examined almost 30 years ago.
In the part " animal experiments" :
1/ the text ought to present/comment figure 5 and 6 - the text ought to be coherent with the description of figure 5 and 6; do not introduce additional knowledge beyond the text - no surprises beyond the text. The text ought to comprise everything.
2/ you did not present the method of this part of study: number of rats - how many of them : 7+8+8 (written down in 161-162)?
There are too many "shortcuts" or maybe the text would be better understood if you would describe materials and methods earlier then results.
Finally:
- there are many "(Error! Reference source not found) :))
- in 21-22, I propose: "... a very close relation between the proportion of pro-inflammatory macrophages and LDL cholesterol levels.." You have just doubled " proportion"
- in 212 : you wrote " cholesterol depletion might may disrupt " - I propose: might disrupt
- in 239 : "information" ought to be
- in 305: instead of "may have led" I propose: might lead
- in 342-343: "The gating strategy is shown in Error!" - ?
- in 194: "....in animals and, also, in ..." Omit one comma, please - I propose the last one
Author Response
I propose you the order: Introduction, Materials and Methods, Results, Discussion, and Conclusions (optional).
I agree that, at least the Methods section, should precede the Results section (the order has been changed in the manuscript accordingly).
I have expanded the description of the study population. Moreover, I have added the descriptions and comparisons of the groups using versus not using statins, as suggested by the other reviewer. If you do not consider it enough, please let me know more specifically.
Although we pooled two groups, from two separate studies, subject enrolment was taking place continually except for a short period. The methods for adipose tissue processing and analysis were the same. As mentioned in the manuscript, the only difference between the study groups was found in age. There were no differences in means of macrophage populations and even in any other analysed parameters. Moreover, the earlier study focused on human adipose tissue macrophages in relationship to cardiovascular risk factors and the later study focused on the relationship between adipose tissue macrophages and macrophages present in the arterial wall (mainly focused on macrophages in the renal artery and perivascular adipose tissue surrounding the artery but also visceral adipose tissue from outside of Gerota´s fascia).
You did not specify clearly the investigated subgroups, e.g. (line 84-86) " 11 (9.3%) subjects receiving statin therapy and 44 (37.61%) subjects with hypercholesterolemia defined as total cholesterol > 5 mmol/l and/or use of lipid lowering agents" - which one? and moreover define clearly please, the sort of groups. How to understand these numbers 11+ 44 ? - it does not give us 118. What about the rest of subjects?
The whole enrolled group (n=118) included 11 subjects on statin treatment and 44 subjects with hypercholesterolaemia. These subgroups (statin treatment, hypercholesterolaemia) were compared to healthy subjects of the rest of the whole group without cardiovascular risk factors followed in this study.
Do not consider the well known facts like the relation between statins and lipids in the "chapter" presenting "animal experiments" - 123-146. Remove this part, please. Be focused on macrophages, please. The anti-inflammatory effect of statins is also well known but still is not highlighted adequately.
Most probably you added this part of paper (animal experiments) due to the results concerning macrophages in rats. Your paper has a value because of the anti-inflammatory effect of statins on macrophages not because you analyzed something (lipids) which had been examined almost 30 years ago.
We are sure that the hypolipidemic effect of statins has been repeatedly documented. We originally included lipoprotein data of our specific genetic rat model but, according to your suggestion, data were reduced. The data in Figure 4 describing detailed analysis of lipoprotein fractions in our animal model were excluded and the effect of statin on plasma lipids was described in a shortened paragraph in the Methods section. Because of this specific rat model, we feel this information is necessary for understanding.
In the part " animal experiments" :
1/ the text ought to present/comment figure 5 and 6 - the text ought to be coherent with the description of figure 5 and 6; do not introduce additional knowledge beyond the text - no surprises beyond the text. The text ought to comprise everything.
We have included additional information so the text is coherent with the description of Figures 5 and 6.
2/ you did not present the method of this part of study: number of rats - how many of them : 7+8+8 (written down in 161-162)?
I have added the numbers for individual groups to the manuscript – controls (n=7), high cholesterol diet only (n=8), high cholesterol diet plus statin treatment (n=8).
There are too many "shortcuts" or maybe the text would be better understood if you would describe materials and methods earlier then results.
I have removed unnecessary shortcuts.
Finally:
- there are many "(Error! Reference source not found) :))
- in 21-22, I propose: "... a very close relation between the proportion of pro-inflammatory macrophages and LDL cholesterol levels.." You have just doubled " proportion"
- in 212 : you wrote " cholesterol depletion might may disrupt " - I propose: might disrupt
- in 239 : "information" ought to be
- in 305: instead of "may have led" I propose: might lead
- in 342-343: "The gating strategy is shown in Error!" - ?
- in 194: "....in animals and, also, in ..." Omit one comma, please - I propose the last one
I really apologise for the “Error!...” I did not check the uploaded manuscript properly. I have solved the problem, and agree with your other comments.
The language will be proofed by MDPI language editors
Reviewer 2 Report
Comments:
The article reports sort of another pleiotropic effect of the statin drug class and or associations between cholesterol and inflammatory state of macrophages. The authors refer to their previously published work showing a decrease of proinflammatory macrophages in visceral adipose tissue in humans. Here they demonstrate an independent association between LDL Cholesterol and the proportion of proinflammatory macs. Also, a direct anti-inflammatory effect of Fluvastatin on macrophage polarization was seen. The authors speculate on the mechanism. They went on concluding that statins may convey an anti inflammatory effect on organism level.
Comments:
Whilst the conclusions of this manuscript are not earth shattering, they are largely in line with previous literature. The novelty is of limited significance and yet, the findings may be worth publication, but the article has several shortcomings that would have to be improved/modified before publication could be considered. The authors should also be are fairly conservative in the interpretation of their data. They may indicate even the need for study of larger cohorts of patients
In detail:
-The study effects may be caused by the effect od administration of the statin (pleiotropic anti-inflammatory) or the change in LDL cholesterol. Could also be a consequent change in oxLDL chol.
-Pooling the data of the independent studies is inappropriate. One could also criticize that the results should be part of the fist paper. It is also inappropriate to publish associated results in fractions to increase number of publications. Thus, the number of study subjects for the current study must either be increased , or better, the current results should be confirmed by recruiting a validation cohort.
-The association of the polarization and the LDL-c is of moderate significance. Thus, it should be listed what patient received what statin in what dosage for what time in addition to any possible co-medication in order to have a detailed overview on the study population. Lipoprotein frations should be shown. oxLDL should be measured
-It should be mentioned from what part the visceral adipose section the material was obtained. Table 1 is useless. It should provide the clinicolaboratory data of subjects on statin and the controls in comparison. What was the indication for administration of the appropriate statin? Anti-inflammatory markers in peripheral blood were measured. Was there a difference in between the groups.
-Lipoprotein fractions should be presented from the rat study. What is the rationale for the use of fluvatatin? A newer statin may have been a better choice, with regards to the clinical relevance and with regards to effect. In rodents, the HDL is the predominant lipoprotein class. When the effects presented are valid in rodetns as well, it may not at all be solely dependent on LDL cholesterol (but on the statin?, HDL ? anything else)
-The differentiated macrophages should be tested for the functional capacity to build foam cells, e.g. by uptake of diloxLDL.
-Figure 2: absolute changes should be shown, pecentages are misleading.
- The application of higher grad stochastic methods in cohort of this size bears the danger of model overfitting. A power calculation should have been performed based on the results of the previous study in order to avoid laborious stochastics for the analysis.
Figure 5: the results presented in figure 5 are of borderline significance, if at all.
Fig 6: NO measurement is certainly only a start and totally insufficient
-Discussion is way to long and can be shortened by at least 50%
Author Response
In detail:
-The study effects may be caused by the effect of administration of the statin (pleiotropic anti-inflammatory) or the change in LDL cholesterol. Could also be a consequent change in oxLDL chol.
Based on our data we cannot be sure about the exact pathway of statins on macrophage polarisation. However, the results of our in vitro experiments suggest a possible direct influence of statins at least at the metabolic status of macrophages.
-Pooling the data of the independent studies is inappropriate. One could also criticize that the results should be part of the fist paper. It is also inappropriate to publish associated results in fractions to increase number of publications. Thus, the number of study subjects for the current study must either be increased, or better, the current results should be confirmed by recruiting a validation cohort.
Both studies were conducted with the same methodology using the same technique of measurement of the main parameters. Therefore, it was justified to pool the data from both studies and perform their analysis. All 118 LKDs were being included continuously from 2013 to 2019 except for a short period.
The first study took place from 2013 to 2015, its main objective being to determine the effect of cardiovascular risk predictors on macrophage polarisation in fat tissue. The findings have been published. They also inspired us to start a new project (2016-2019) with the objective to determine if there is a relation between macrophage polarisation in the arterial wall and in perivascular fat tissue. In this project, visceral fat tissue was also analysed in order to determine if the influence of fat tissue is local or systemic. Since the analysis methods were identical in both projects, it was possible to pool data from the first and second studies as LKD enrolment was identical as were adipose tissue processing and labelling.
We also have put the Methods section before Results section to make it more understandable.
-The association of the polarization and the LDL-c is of moderate significance. Thus, it should be listed what patient received what statin in what dosage for what time in addition to any possible co-medication in order to have a detailed overview on the study population. Lipoprotein frations should be shown. oxLDL should be measured
Regarding statistical significance, we assume that, given the heterogeneity of our group of heathy individuals, the significance of the relationship of the proportion of pro-inflammatorily polarised macrophages to LDL concentration is rather strong (p<0.01) and the correlation of anti-inflammatory macrophages to LDL is even stronger (p<0.001).
The lipoprotein fractions (at least total cholesterol, triglycerides, LDL and HDL) are shown in the manuscript. The analysed group is compared to the general Czech population slightly healthier as mentioned in the manuscript by Poledne et al. J Lipid Res 2016.
-It should be mentioned from what part the visceral adipose section the material was obtained. Table 1 is useless. It should provide the clinicolaboratory data of subjects on statin and the controls in comparison. What was the indication for administration of the appropriate statin? Anti-inflammatory markers in peripheral blood were measured. Was there a difference in between the groups.
Visceral adipose tissue was obtained during hand-assisted nephrectomy of LKDs from outside of Gerota´s fascia during cleaning of the kidney prior to transplantation as mentioned in the Methods section. The clinicolaboratory characteristics of subjects on statin therapy and controls have been added to Table 1. Thank you for your helpful comment.
All subjects on statin therapy were receiving statins for at least 2 years prior to transplantation, presumably because of hypercholesterolaemia (prescribed by their physician) [rosuvastatin (n=1), atorvastatin (n=7); one subject had a combination of rosuvastatin and atorvastatin (n=1), simvastatin (n=1), and fluvastatin (n=1)].
The concentrations of selected cytokines measured in blood by Luminex assay (TNF-α, IL-1β, adiponectin, IL-6, IL-10, MCP‑1) showed no differences between the groups with and without statins.
-Lipoprotein fractions should be presented from the rat study. What is the rationale for the use of fluvatatin? A newer statin may have been a better choice, with regards to the clinical relevance and with regards to effect. In rodents, the HDL is the predominant lipoprotein class. When the effects presented are valid in rodetns as well, it may not at all be solely dependent on LDL cholesterol (but on the statin?, HDL ? anything else)
Fluvastatin was used only for in vitro experiments due to its water solubility (the only statin) making it possible to simply add it to the culture media. If using any other statin dissolved in a lipophilic solvent, we would need to administer the solvent into all experimental wells, which could influence the cells as such. In our animal model, we used simvastatin (5mg/kg body weight/day) as mentioned in the Methods section. It is necessary to stress, that in our genetic model of hypercholesterolaemia, HDL particles are far from being dominant. The effect of the statin to the VLDL, IDL and LDL fractions was presented in the original version of our manuscript but the other Reviewer required to withdraw this part.
-The differentiated macrophages should be tested for the functional capacity to build foam cells, e.g. by uptake of diloxLDL.
We plan to carry on with our in vitro experiments analysing in detail the exact mechanism of the statin effect on macrophage polarisation using also isolated tissue macrophages. We will analyse the influence of statins on the production different cytokines and other metabolites important for their functions including, for example, cholesterol efflux. It would be also good include function tests (as mentioned by you) such as oxLDL uptake and, also, foam cell formation capacity. Thank you for your helpful suggestion.
-Figure 2: absolute changes should be shown, pecentages are misleading.
The subsets of macrophages were expressed as relative number (in percentage). Therefore, the differences (changes) were absolute and expressed as percentage, i.e., the difference of absolute numbers of subsets.
We have changed legend to Figure 2 (now Figure 3) to be better understandable. Now it is formulated as “Data are expressed as the difference in the relative numbers of pro‑inflammatory, transient and anti-inflammatory phenotypes in VAT. “
- The application of higher grad stochastic methods in cohort of this size bears the danger of model overfitting. A power calculation should have been performed based on the results of the previous study in order to avoid laborious stochastics for the analysis.
To avoid bias or model overfitting, the Bayesian approach was applied. Therefore, the power test was not justified – this test does not exist in this approach.
Figure 5: the results presented in figure 5 are of borderline significance, if at all.
I am aware of the rather borderline significance in the analysis, but the differences are still significant. We present the results only in the context of the results of analysis of human adipose tissue macrophages and the results of our in vitro experiment.
Fig 6: NO measurement is certainly only a start and totally insufficient
I agree with you. It is just an indication of the direct effect of statins on macrophage polarisation. Additional experiments including functional experiments are needed (as mentioned above).
Based on very recent results, it seems that fluvastatin in vitro, besides changes in NOS activity, changes even gene expression in the influenced cells towards anti-inflammatory expression. However, these analyses are not final.
-Discussion is way to long and can be shortened by at least 50%
The discussion has been substantially shortened.
The language will be corrected once more by MDPI language editors.
Round 2
Reviewer 2 Report
This reviewer believes that the manuscript has somewhat improved.